# Towards a Resilience to Stress Index Based on Physiological Response: A Machine Learning Approach

**DOI:** 10.3390/s21248293

**Published:** 2021-12-11

**Authors:** Ramon E. Diaz-Ramos, Daniela A. Gomez-Cravioto, Luis A. Trejo, Carlos Figueroa López, Miguel Angel Medina-Pérez

**Affiliations:** 1Department of Computer Science, School of Engineering and Sciences, Campus Monterrey, Tecnologico de Monterrey, Monterrey 64849, Mexico; diazramo@ualberta.ca (R.E.D.-R.); a01181520@exatec.tec.mx (D.A.G.-C.); 2Department of Computer Science, School of Engineering and Sciences, Campus Estado de México, Tecnologico de Monterrey, Atizapán 52926, Mexico; migue@tec.mx; 3Department of Psychology, School of Health, Campus Ciudad de México, Tecnologico de Monterrey, Ciudad de México 14380, Mexico; carlos.figueroa@tec.mx; 4Altair Management Consultants, Calle de José Ortega y Gasset 22-24, 5th Floor, 28006 Madrid, Spain

**Keywords:** resilience to stress, physiological response, machine learning, clustering

## Abstract

This study proposes a new index to measure the resilience of an individual to stress, based on the changes of specific physiological variables. These variables include electromyography, which is the muscle response, blood volume pulse, breathing rate, peripheral temperature, and skin conductance. We measured the data with a biofeedback device from 71 individuals subjected to a 10-min psychophysiological stress test. The data exploration revealed that features’ variability among test phases could be observed in a two-dimensional space with Principal Components Analysis (PCA). In this work, we demonstrate that the values of each feature within a phase are well organized in clusters. The new index we propose, Resilience to Stress Index (RSI), is based on this observation. To compute the index, we used non-supervised machine learning methods to calculate the inter-cluster distances, specifically using the following four methods: Euclidean Distance of PCA, Mahalanobis Distance, Cluster Validity Index Distance, and Euclidean Distance of Kernel PCA. While there was no statistically significant difference (p>0.01) among the methods, we recommend using Mahalanobis, since this method provides higher monotonic association with the Resilience in Mexicans (RESI-M) scale. Results are encouraging since we demonstrated that the computation of a reliable RSI is possible. To validate the new index, we undertook two tasks: a comparison of the RSI against the RESI-M, and a Spearman correlation between phases one and five to determine if the behavior is resilient or not. The computation of the RSI of an individual has a broader scope in mind, and it is to understand and to support mental health. The benefits of having a metric that measures resilience to stress are multiple; for instance, to the extent that individuals can track their resilience to stress, they can improve their everyday life.

## 1. Introduction

Stress has become a primary topic in the pursuit of mental health for modern society. The main reason for this is that stressful events can act as a precursor to major psychiatric conditions, such as anxiety and depression [1]. Consequently, these mental illnesses can lead to physical problems, such as an increase in the risk for cardiovascular diseases [1,2]. Given the mental and physical problems that stress can cause, there has been an increased interest in studying ways to timely detect and prevent stress in individuals [3,4,5,6].

Although multiple works have examined stress detection [1,2,3,4,5,6], we have identified that the variables explored in these studies have not measured resilience to the stress of an individual [3,4,5,6]. Ćosić et al. [7] identified that there is no direct quantifiable index to monitor the degree of resilience to stress of an individual based on physiological responses.

Resilience to stress is defined as the properties contributing to the speed and amount of possible recovery of physiological variables after exposure to a stressful event [8]. Since it is challenging to avoid stressful situations in a fast-paced environment, it has become increasingly important for individuals to develop a mindset to overcome stress [8,9,10].

Many psychophysiological tools measure resilience based on questionnaires. These studies assign a scale depending on the individual’s response to the questions. The Connor-Davidson Resilience Scale (CD-RISC) [11], developed in 1999, the Resilience Scale for Adults (RSA), developed in 2001 [12], and the Resilience in Mexicans (RESI-M) built in 2010 [13] are examples of these studies. These tools assign a scale based on how the subjects have felt in the previous weeks. The variety of these scales gives psychologists and psychiatrists a wide range of tools to measure resilience, considering features that may reflect resilience’s underpinnings. Some of the features are adaptability when coping with stress, strong self-esteem/confidence, adaptability when coping with change, social problem-solving skills, humor in the face of stress, strengthening the effect of stress on others. These tools are focused on measuring the full spectrum of resilience, but they do not provide resilience to stress, specifically.

This study adds to the application of unsupervised machine learning by defining an appropriate methodology to quantify the alteration of physiological features. The research focuses on cluster distances to quantify the alteration of five physiological features. To accomplish this, we first applied Principal Components Analysis (PCA) to the data and observed that features’ variability among test phases can be depicted in a two-dimensional space; in such a space, the values of each feature within a phase are well organized in clusters. In order to derive a new resilience index, we used non-supervised machine learning methods to calculate the inter-cluster distances, specifically using the following four methods: Euclidean Distance of PCA, Mahalanobis Distance, Cluster Validity Index Distance, and Euclidean Distance of Kernel PCA. Additionally, this study provides a different approach to combine static and dynamic features by transforming the dynamic features, in this case, a time series, without vectorizing sequential data or using an ensemble method for conventional classification models. Results are encouraging since we demonstrate that the computation of a reliable resilience index is possible. The main contributions of this paper are the following:It proposes a new index that quantifies resilience of an individual to stress based on physiological response.It proposes a new factor that quantifies the alteration intensity of the physiological features.It validates the correctness of the resilience to stress index.

The organization of this paper is as follows: Section 2 presents related work to ours, focused on physiological features and methods mainly used by the research community. Section 3 describes the dataset, the methodology used for signal processing, and four inter-cluster distance algorithms to calculate the index of interest. Next, Section 4 presents the new resilience to stress index and the alteration factor, followed by Section 5 which describes the index validation. Lastly, Section 6 presents our conclusions, limitations, and future work.

## 2. Related Work

The following literature review concerning stress assessment with physiological sensors was undertaken to highlight physiological variables related to psychological stress.

The first related work of interest is the one from Healey and Picard [3]. In this study, the authors analyzed physiological variables during real-world driving environments to determine a driver’s relative stress level. Electrocardiogram, electromyogram, skin conductance, and respiration were monitored during five-minute intervals. This enabled identifying three distinct levels of psychological stress with an accuracy of 97%. The researchers concluded that skin conductivity and heart rate were highly correlated with stress levels prediction. They assessed stress level with a questionnaire and video coding. The questionnaire was based on the relative sentiment of the subject, where he/she specified the level of stress. The video coding evaluated the movement of the subject and assigned a score depending on the number of tasks associated with it. The questionnaire results were analyzed with ANOVA and determined that the means were significantly different with 95% confidence (*p*-value > 0.001).

A second study of interest performed by Ćosić et al. [7] evaluates stress resilience based on similar physiological features during the selection of air traffic controllers (ATCs). During the study, the authors measured the variability of each feature by using traditional statistical techniques while assessing the differences between a control and a stress-induced group. The input-induced stimulation included the simulation of different conditions that an ATC is prone to encounter. Their results reported statistically significant differences (*p* < 0.05) between the groups. This research focused on eight relevant physiological features as tools for stress resilience assessment. Additionally, they applied psychological questionnaires relevant for resilience assessment such as the Connor-Davidson Resilience Scale, Anxiety Sensitivity Index, and Core Self-Evaluations Scale. Nevertheless, the authors do not analyze the relationship between the index and the psychological questionnaires. Instead, they compared it against a control sample of students against ATC candidates, where they concluded that the experimental group of ATC candidates are more resilient than the control group, according to their stress resilience assessment. While this study provided very insightful information for measuring individual resilience to stress, the study was performed using conventional statistics.

An improvement to the same ATC problem was reported by Šarlija et al. [14]. Their approach included analysis of electrocardiography, electromyography, electrodermal activity and respiration for assessment of physiological features of stress resilience. They applied a two-class classifier to evaluate the predictive power of the selected features, and classified a candidate as either high or low performing on an ATC simulator. They yielded a 78.16% classification accuracy.

A final relevant study is the one from Lu, Wang and You [15]. This study identified resilience based on CD-RISC traits, and studied physiological recovery to stress. Subjects were exposed to stress in a protocol with seven stages: baseline, stress anticipation 1, stress 1, post-stress 1, stress anticipation 2, stress 2, and post-stress 2. The mental stress was induced with the Trier Social Stress Test [16]. The results indicate that high-trait-resilient participants have a complete recovery from the first and second stress of the systolic and diastolic blood pressure compared to low-trait-resilient subjects. The study concluded that an adaptive physiological response pattern to recurrent stress is found in high-trait-resilient individuals based on electrocardiogram data [15].

## 3. Methodology

This section describes the dataset, the employed methodology and the non-supervised learning techniques used to measure resilience to stress based on physiological response. Moreover, in this section we explore different approaches to transform dynamic data and propose four methods to compute the Resilience to Stress Index (RSI). Figure 1 shows the flowchart of the process to compute the new metrics and it is explained in detail in the following sections.

### 3.1. Dataset

The dataset used for this study is a collection of physiological features from 71 individuals from Tecnólogico de Monterrey University in Mexico. It has a gender distribution of 37 males and 34 females. The age of the participants ranges from 18 to 28 years old. The subjects were excluded if they had a cardiovascular disease that could impact the heart rate variable.

For each of the subjects, five variables were measured using a biofeedback device. The biofeedback device has a sampling rate of 256 observations per second from a continuous signal. This rate allowed the five sensors (even slow signals, with no high-frequency component) to have sampling rates without precision loss. The analog signal was processed by the encoder (ProComp Infiniti) and finally sent to the computer.

The five variables measured are skin conductance, blood volume pulse, peripheral temperature, electromyography, and breathing rate. These variables were assessed with analog sensors and captured by the analog-digital converter of the biofeedback equipment. The measurement obtained from each variable is described in Table 1; the variables are described in detail in the following section.

Figure 2 illustrates data of a single subject during the psychophysiological stress test. The five features are represented in each slot through time (256 samples per second). We can observe from the figure the presence of clear spikes in electromyography, blood volume pulse, and breathing signals. This indicates that the signals need to be filtered and preprocessed before the machine learning model experimentation.

### 3.2. Physiological Signals

This section introduces the physiological features examined in this work, previously linked with psychological stress and resilience to stress.

To measure physiological changes in the body of the subjects under study, we employed the biofeedback device ProComp5 Infiniti System w-BioGraph Infiniti Software T7525 (https://thoughttechnology.com/procomp5-infiniti-system-w-biograph-infiniti-software-t7525 (accessed on 6 December 2021)). By means of this device, we measured five different body responses: peripheral temperature, electromyography, breathing rate, blood volume pulse, and skin conductance. The variables were selected as they have previously been identified to suffer an alteration while an individual is experiencing stress. Next, we present a brief background of each physiological feature followed by the sensor description.

#### 3.2.1. Skin Conductance (SC)

The study led by Harker [17] concluded that psychological sweating in response to stress, anxiety, and pain occurs over the whole body. However, it is more evident on the palms, soles, face, and axilla. Moreover, Ling et al. [18] reached a similar conclusion. They induced stress with mental arithmetic testing and similar procedures of measuring relaxation and testing phases. They concluded that stress is positively correlated with skin conductance.

The theory of galvanic skin response analysis is based on the assumption that skin resistance varies with the state of the skin’s sweat glands. The sweating of the human body is regulated by the Autonomous Nervous System. In particular, if the sympathetic branch of the Autonomic Nervous System is highly excited, sweat gland activity also increases, which in turn increases skin conductance, and vice versa.

The skin conductance sensor measures the skin’s ability to conduct electricity. The device determines skin conductance as changes in an electrical current occur [19]. The signal is encoded with the biofeedback device and captured in the computer. A micro-Siemens unit is equivalent to a unit of the inverse of mega-ohms. The principle is to apply a small current through two electrodes, usually strapped to two fingers of one hand, and measure its conductance variation. In our setup, we strapped the sensor’s electrodes to the index and ring fingers of the right hand. As a person reacts to stress, the skin’s conductance increases, as the individual tends to sweat [18].

#### 3.2.2. Peripheral Temperature (PT)

A change in body temperature is experienced in different parts of the body when exposed to stress. Vinkers et al. [20] studied the effects of stress on the core and peripheral body temperatures. The authors concluded that stress exposure decreases the temperature of the intestines and specific skin locations, such as fingertips and the base of the finger. Vinkers et al. precluded previous studies [21], where they indicated that temperature uniformly rises in response to stress. Hence, we can expect that exposure to stress lowers the peripheral temperature (fingertip). The study analyzed the data with typical statistics techniques, specifically ANOVA.

The sensor from the biofeedback converts changes in temperature to electrical current changes to be encoded by the device module. The sensor is placed in the palm of any finger and varies according to the amount of blood flowing through the skin. In our case, we strapped the sensor to the palmar side of the index finger of the left hand. As a person reacts to stress, their fingers tend to get colder. Usually, peripheral temperature changes are moderately slow.

#### 3.2.3. Blood Volume Pulse (BVP)

Heart Rate Variability (HRV) can be obtained by non-invasive methods and allows evaluating the action of stress on the body’s physiology. It is defined as the variation that occurs in the time interval between beats when the body copes with various situations. Its behavior is conditioned by the inspiratory and expiratory breath processes and depends on the autonomic modulation of the heart reflecting vagal, sympathetic, parasympathetic activity, and circadian rhythms in response to mental stress. There are some factors that decrease HRV such as depression, physical, and mental stress [22]. More recent studies found HRV was significantly lower in stress conditions in comparison with basal and recovery states [23,24].

The sensor principle is to bounce infrared light against the skin surface and measure the amount of reflected light. The amount of blood present in the skin varies accordingly with the heartbeat. When a heart pulse occurs, there is more blood present in the skin, blood reflects red light, and consequently, more light is reflected. Between pulses, the amount of blood in the veins decreases, and more red light is absorbed, then the amount of light returning to the sensor is lower [25].

The blood volume pulse is a relative measure; it does not have a standard unit. The typical signal shows a substantial rise with the systolic contraction, followed by a slower fall. Usually, with changes in sympathetic arousal, the peak-to-peak intensity of the signal can increase and decrease. From the BVP signal, the device software can calculate heart rate, inter-beat interval, and HRV. In our setup, we placed the sensor on the palmar surface of the thumb fingertip at the right hand.

#### 3.2.4. Breathing Rate (BR)

The breathing rate has been of interest, as it increases when an individual is induced to mental stress [26,27]. The study of Masaoka and Homma [28] concluded that unpleasant emotions alter breathing patterns. The study induced mental stress by amplifying noise using sounds from the environment. They were assuming that noisiness increases with the loudness of the induced sound. These relationships of noise and stress were previously studied by Berglund [29] in 1976. For the induced test, during the procedure each subject was consistently tracked twice during three conditions: baseline of resting-state, three minutes; physical load-mental stress, two minutes; and stress, three minutes. The relationships between the control and the tests were investigated using a repeated-measure analysis of variance (ANOVA).

The respiration sensor from the biofeedback is stretch-sensitive. It can translate the expansion and contraction of the rib cage or abdominal area to an increase and fall of the signal while strapped across a person’s chest or abdomen. A relative indicator of chest expansion is the respiration signal. The biofeedback equipment does not produce standard measurement units for respiration. From the raw signal, the device software is able to calculate the respiration rate and the relative breath amplitude.

Several breathing patterns can be identified, but the respiration signal has no standard waveform. There is normally a quick rise that slows at the top of the breath, followed by a rapid decline near the end of the breath. When the participant focuses on a task or speaks, the breathing rhythm will change itself. In our setup, we placed the sensor in the subject’s abdomen.

#### 3.2.5. Electromyography (EMG)

Ludenberg et al. [30] conducted a study of the biological reaction to stress of the body. The results concluded that the induced stress session significantly increased systolic and diastolic blood pressure, heart rate, urinary catecholamines, salivary cortisol, and electromyography activity compared to the baseline trapezius muscle (indicating muscular tension in the trapezius area). The induced stressor consisted of mental arithmetic, the Stroop color-word test, the cold pressor test, and standardized test contractions. This research supports the theory that psychological stress contributes to musculoskeletal disorders by enhancing muscular tension in both low-load and no-load situations. A more recent study by Hermens et al. [31] conducted similar experiments in 2014 and reached the same conclusion.

The sensors in the biofeedback device measure muscle activity by detecting the electrical impulses generated by muscle fibers when they contract. Since all muscle fibers contract at various speeds within the sensor recording area, the signal continuously changes the potential difference between its positive and negative electrodes. The amount of muscle fibers recruited during any given contraction, depends on the force needed for the motion to be done. For this effect, the power (amplitude) of the resulting electrical signal is proportional to the contraction frequency [32]. For this physiological response, the EMG activity is recorded from the subject’s trapezius muscle.

#### 3.2.6. Final Comments

As per the previous studies, we have identified that the expected variables to increase when an individual is exposed to stress are: electromyography, skin conductance, and breathing rate. Moreover, the variables expected to decrease are peripheral temperature, and heart rate variability. Given this, the time a person takes to return its heart rate variability, respiratory rate, muscle tension, temperature, and skin conductance to its baseline parameters, can help us define the resilience to stress of a person.

### 3.3. Psychophysiological Stress Test

The psychophysiological stress test is a protocol to measure autonomic reactivity and recovery to psychological stress through a psychophysiological mental stress profile. The test includes the simultaneous recording of five physiological responses (peripheral temperature, blood volume pulse, respiratory rate, muscle contraction, and the galvanic response of the skin).

#### 3.3.1. Autonomic Reactivity to Stress

The conceptual definition of autonomic reactivity to stress is the magnitude of physiological activation and recovery responses controlled by the autonomous nervous system in the presence of stressful situations [33]. The operational definition is that the autonomic reactivity to stress will be measured after the magnitude of responses from the sympathetic nervous system to discrete environmental stressors. It considers the changes that are produced by the following physiological responses: galvanic response of the skin, systolic and diastolic blood pressure, heart rate, heart rate variability, and respiratory chest rate. To measure these variables, we employ the Computerized Biological Feedback equipment.

#### 3.3.2. Autonomous Recovery from Stress

The conceptual definition of the autonomous recovery from stress is the measurement of a set of responses inhibited by the autonomous nervous system in its parasympathetic branch through relaxation [33]. The operational definition is that the autonomous recovery from stress is measured from the degree of recovery of basal levels within a previously established period after applying a stressor. We focus on changes produced by the same physiological responses as mentioned above.

#### 3.3.3. Protocol Description

Stress is induced by challenging the subject with mathematical operations and emotional tasks. During the test, the evaluator needs to guarantee a constant pace and obtain quick and correct answers without any distractions such as laughing or moving excessively.

The protocol consists of ten minutes covering five phases; each phase has a total duration of two minutes. There are two stages intended to induce external stressors, two recovery phases after the stressor and one to build a baseline. The first phase consists of the baseline or calibration phase; the evaluator does not give instructions besides that the subject must remain seated and relaxed. This phase allows recording the physiological baseline of each subject. Before starting the second phase, the evaluator gives the instructions for the mathematical and emotional tasks. The induced stressor is applied in phases 2 and 4; phases 3 and 5 are recovery periods.

During the mathematical stressor, the subject must subtract 9 from a three-digit number, as fast as possible (e.g., subtracting 9 from the number 246). Throughout the emotional stressor, the subject must identify a very stressful event that occurred during the last three months, then provide details of the event such as place, day, time, people involved, and recall facts that lead to the most stressful and distressing moment. The protocol is illustrated in Figure 3 and Figure 4 shows the stress test setup.

### 3.4. Signal Preprocessing

This section describes the data preprocessing of the subject’s data obtained using the biofeedback device.

#### 3.4.1. Median Filter

As shown in Figure 2, the signal must be preprocessed before using the data to avoid pseudo detection of peaks or noise. Hence, a median filtering technique is applied to remove peaks and noise from the signal. The kernel size of the filter was selected according to Equation (Equation 1) [34].
(1)$$w=\frac{1}{4f_{s}}*length\left(v\right)$$
where $f_{s}$ is the sampling frequency and length(v) is the total number of instances. The kernel size must be odd as it selects its neighbors to get the median. Hence, a window size of 151 was selected. Additionally, the raw signals from the biofeedback device have an offset at the beginning of the sensor measurements. Therefore, for this analysis, the first 0.5 s of data were removed.

#### 3.4.2. Standard Scaler

The unit of measure of each sensor is on a different scale; therefore, a critical step is standardization. For this, a standard scaler was applied after filtering the data. Equation (Equation 2) is used to standardize each point of all features. Standard scaling is a way of normalizing features by deleting their mean and scaling their variance to one. Since the normalized value is determined solely by the mean and variance, it has some advantages, including being linear, reversible, rapid, and highly scalable [35].
(2)$$z=\frac{x-a}{s}$$
where *a* is the mean of the feature, and *s* is the standard deviation of the variable.

#### 3.4.3. Preprocessing Result

We can observe in Figure 5 that peak points in blood volume pulse and electromyography signals are reduced. Moreover, it illustrates the time division of the corresponding phase during the psychophysiological stress test. Phase 1 corresponds to the first 30,720 instances (first 2 min). At the beginning of phase 2, an external stressor is applied to the subject. Each variable suffers alterations in the second phase in comparison to the signals in the first phase. These signal changes between phases are critical to capture the severity of stress that the individual suffers through the test. Therefore, we can quantify the alteration of these variables using different techniques explained in the next section.

It is important to notice that the signal offset is not an issue in calculating the RSI or the Alteration Factor (AF), because the distances are relative to the first phase (baseline) of each subject. The AF metric is later introduced in Section 4.2.

### 3.5. PCA Visualization

To understand the behavior of the five signals, we can transform the data with the principal components technique and visualize it in a bi-dimensional graph. PCA allows reducing high-dimensional data to a smaller number of dimensions while maintaining the information [36]. Several steps were taken to ensure data comply with PCA assumptions (e.g., multicollinearity [37]). For visual representation purposes, the sample rate was reduced from 256 samples per second to one sample per second. The median of 256 samples per second was calculated to reduce the number of points; this allows us to avoid excessive points in the PCA plot. Next, to increase linear relationship among features, we transformed the data with a power transformation. To accomplish this, we applied a power transformer with the Yeo-Johnson method of the ScikitLearn module (https://scikit-learn.org/stable/modules/generated/sklearn.preprocessing.PowerTransformer.html (accessed on 6 December 2021)).

Now, after scaling and transforming, PCA can be applied to the data. After fitting PCA for all subjects, we obtained a mean explained variance of 0.6384 for two components, which is enough to understand the data behavior visually. Figure 6 illustrates the resulting plot of bi-dimensional space. The five physiological features of these subjects do not have a normal distribution. In the PCA graph, we can observe that each point consists of a second with two dimensions in the first principal component (PC1) and the second principal component (PC2). The color defines the corresponding phase of the stress test. There is an evident displacement between phases, meaning that PCA could capture the behavior of all variables in a two-dimensional space. This behavior in two-dimensional space can be extrapolated to an *n*-dimensional space with the five measured variables, but a two-dimensional space was created for behavior visualization purposes. Hence, we can measure the relative distance between centroids points among phases using all variables.

The preprocessed signals of a subject representing its complete profile were previously shown in Figure 5. After 120 s (30,720 samples), the signals suffer a drastic change. The first phase consists of a baseline where the individual is calm. After 120 s, phase 2 begins inducing external mental stress with analytical problems. It is clear the displacement of values at the beginning of phase 2, meaning the subject is suffering a relative change in its variables. These variations are reflected in the PCA two-dimensional space (Figure 6).

The relative displacement between phases can be compared to the baseline phase (phase 1). It is essential to notice that an *n*-dimensional space with 256 samples per second, instead of one sample per second, allows us to avoid losing information when calculating relative distances between phases. Additionally, the series are time-dependent, and to capture this dependency, it is needed to add non-time dependencies to the dataset before calculating centroids.

#### Additional Features

Time series hold valuable information that varies continuously through time. A first-order difference can help to extract information in the temporal dependence of the series. Differencing is performed by subtracting the previous observation from the current instance. Equation (Equation 3) shows the basic principle to extract additional information by reducing trend and seasonality.
(3)difference(t)=instance(t)−instance(t−1)

For this study, one order difference and an additional differencing in t−2, shown in Equation (Equation 4), were added to capture additional information to calculate distances between phases in an *n*-dimensional space.
(4)difference(t)=instance(t)−instance(t−2)

The resulting dataset consists of 15 attributes for each subject, two additional variables per physiological feature.

### 3.6. Resilience to Stress Index (RSI) Proposal

Now that the information is complete and preprocessed, it can be used to quantify an alteration factor (AF) and the RSI. This work proposes four different approaches to obtain the RSI and the AF values. The common ground of these approaches is that changes in signals due to stress can be compared against a calibration stage. The calibration stage is unique to each subject; hence, all computed distances are relative to this starting point or baseline. The calibration stage is phase 1 of the stress test, where the subject is relaxed and resting for two minutes. We compute the distance between each phase to the calibration phase. In other words, we calculate the distance of phase one against all other phases; that is, $phase_{1}\to phase_{i}$, where *i*
∈{2,3,4,5}.

Next, we explain our approach to compute distances between phases. The primary assumption is to consider a phase as a cluster of points (physiological measurements) close to each other. We then use four well-known techniques to calculate the distance among clusters: Euclidean distance of PCA and Kernel PCA, Mahalanobis distance, and Cluster Validity Index distance.

#### 3.6.1. Euclidean Distance of PCA Components to Calculate Inter-Phase Distances

PCA uses a covariance matrix which eliminates multicollinearity between attributes [38]; therefore, euclidean distance can be used to calculate the distance between clusters. PCA was constructed for all subjects using the same preprocessing and assumptions explained in Section 3.4. The mean average of the explained variance ratio with 15 components was 100% with a standard deviation of zero. With 14 components, the explained variance ratio slightly decreases to a mean value of 99.87% with a standard deviation of 0.11. In this study, to minimize information loss, we decided to use 15 principal components to compute the distances between clusters.

The relative displacement between phases is now compared to the calibration phase (phase 1). It is crucial to notice that an *n*-dimensional space (n=15 in our case) with 256 samples per second minimizes information loss when calculating relative distances between phases. Hence, the centroids are calculated using 30,720 observations per phase (a total of 153,600 observations per subject). The results obtained in this step, include the distances between the centroid of the calibration phase and the subsequent psychophysiological stress test phases.

#### 3.6.2. Mahalanobis Distance to Calculate Inter-Phase Distances

The second technique to measure cluster distance is the Mahalanobis distance [39]. Compared to the euclidean distance, the Mahalanobis distance can be calculated without standardizing the data [40]. It can measure distances between points in multivariate space, even if their variables are correlated [39]. For this study, the distances are calculated using the Scipy Python package.

The objective of the method is to calculate the distance of the pair of phases: $phase_{1}\to phase_{i}$, where *i*
∈{2,3,4,5}, using the following steps:1.Calculate the inverse of the covariance matrix with the instances that correspond to the pair of phases to calculate the distance (e.g., phase 1 and phase 2 instances).2.The centroid of the baseline (phase 1) is calculated using the mean of each feature. Therefore, the resulting centroid will have a vector of 15 attributes.3.Compute the Mahalanobis distance of each instance vector against the centroid vector of the baseline.4.Compute the mean of all calculated distances of the previous step to obtain the result.

#### 3.6.3. Cluster Validity Index Distance to Calculate Inter-Phase Distances

The third approach to measure cluster distance is employing the Silhouette index [41]. The separation metric is based on the nearest neighbor distance, and the Mahalanobis metric is used as the distance metric [42]. The Silhouette score is calculated with the Sklearn Python package. The function returns the coefficient of the mean silhouette for all samples. It calculates the mean of the nearest cluster distance and the mean of the intra-cluster distance.

A Silhouette value of zero indicates that the clusters are overlapping. A negative value usually means that a sample was wrongly assigned and could belong to a different cluster [43]. A value close to one indicates that the clusters are well matched. Since the Silhouette coefficient is a measure of cohesion and separation between clusters [41], all pairs of clusters’ behavior can be calculated.

The resulting dataset consists of the silhouette coefficient of each combination of pairs of the data. For this application, a value closer to one means that the compared pair of phases has fair cohesion and separation between each other, meaning that the subject has phases that do not overlap and the biofeedback signals are different.

#### 3.6.4. Euclidean Distance of Kernel PCA Components to Calculate Inter-Phase Distances

Non-linear methods such as Kernel PCA [44], can compute the principal components of an *n*-dimensional feature space by integrating operators and non-linear kernel functions. The main idea of the Kernel PCA is to map the original space into a feature space with non-linear mapping and later obtain the principal components in that feature space [45].

Similar to the first method, the relative displacement between phases is now compared to the calibration phase, but instead of computing the distances in the linear space, this method avoids transforming the data to comply with linearity and computes the distances in the non-linear mapping feature space. However, as the standard implementation of the Kernel PCA scales poorly with the problem size [46], a sample rate transformation of 1 for 256 samples was applied to minimize the computation time.

The results obtained with this method include the distances between the centroid of the calibration phase and the subsequent psychophysiological stress test phases, similar to the Euclidean distance of PCA components.

#### 3.6.5. Statistical Analysis and Time Complexity

In our study, we used the following statistical methods: to test if the data has a normal distribution, we used the Anderson-Darling test; to test significance among distance metrics methods of non-parametric samples, we used the Friedman test; and to obtain the correlation coefficient, we used Pearson or Spearman.

We first analyzed the normality of the residuals with the Anderson-Darling test. The statistical test was conducted for the four methods (Euclidean of PCA, Mahalanobis, Silhouette with Mahalanobis, and Euclidean of Kernel PCA) and a dataset of 71 paired samples. For this analysis, we used the Python package Autorank [47] and a significance level of 0.01. We rejected the null hypothesis (stating that the population is normally distributed) for the Euclidean of Kernel PCA (p<0.0005) and Silhouette with Mahalanobis (p<0.0005) methods as the *p*-values obtained were smaller than the significance level. Given these results, we used the non-parametric Friedman test to assess the significance between the methods [48].

We failed to reject the null hypothesis (p=0.041) of the Friedman test, concluding that there was no difference in the central tendency of the populations, for all methods. Hence, we conclude that there is no statistically significant difference between the median values of the populations. The results of the median absolute deviation (MAD), the median (MD), and the mean rank (MR) are shown in Table 2.

Regarding the time complexity of the machine learning algorithms used int his work, PCA has a time complexity of $O(n^{3})$, Standard Scaler of O(n), Inter-phases distances of $O(n^{2})$, and Resilience to Stress Index of $O(n^{2})$ (see Section 4.1). Given that the highest time complexity of these components is $O(n^{3})$, we can infer that the time complexity of the proposal is $O(n^{3})$.

## 4. Results

The proposed index can capture the physiological response of an individual. After computing the cluster distances or the silhouette coefficient between pairs, we calculated the RSI and the maximum AF values. The process to obtain these values is explained below.

### 4.1. Resilience to Stress Index (RSI) Calculation

The RSI calculation based on cluster distances consists of measuring the cluster distance between phase four (the last stressor phase) and phase five (the last recovery phase) of an individual. The RSI is defined by the following equation (Equation (Equation 5)).
(5)$$RSI=\frac{\Delta R}{\Delta S}$$
where ΔR is the difference between the centroid of phase four and the centroid of phase five of the subject, and ΔS is the maximum ΔR of the sample.

An RSI value close to one implies that the subject’s physiological responses could quickly recover from the mental stress challenge. On the other hand, negative values indicate that the individual could not recover its physiological state (baseline) after the last induced stressor. After computing the RSI values for all subjects using the four distance metrics discussed previously, we compared the variance between them and proved no significant statistical difference (p<0.01).

### 4.2. Maximum Alteration Factor

To determine the maximum stress level of subjects, we measured their maximum distance from phase 1 to the rest of the phases. We propose a new metric named the alteration factor (*AF*) to measure the stress level of a subject. This metric is obtained by comparing the maximum stretch (distance) of the subject to the maximum stretch of the sample. Equation (Equation 6) shows the calculation of this metric.
(6)$$AF=\frac{SubMaxS-SamMinS}{\Delta SamST}$$

SubMaxS is the maximum stretch of the subject compared to his/her baseline. SamMinS is the minimum stretch of the sample, and ΔSamST is the difference between the maximum and minimum distances of the sample in the dataset. An AF value close to one suggests that the subject’s physiological response has suffered a higher alteration than most individuals in the sample. On the other hand, a value close to zero indicates that the variables did not suffer such a noticeable change with respect to the sample.

### 4.3. Subject Examples Analysis

This section illustrates the results of the metrics created by two subjects with opposite RSI values. The example illustrates the RSI based on the Euclidean distance of the PCA because the RSI with this method allows a visual inspection of a subject’s behavior through the psychophysiological stress test and evaluates its corresponding RSI. Figure 7 depicts the five standardized physiological features of two subjects through the five phases of the psychophysiological stress test. The vertical red line separates each phase through time. Figure 7a shows a positive RSI value for Subject 1, whereas Figure 7b displays data for Subject 2, with a negative RSI.

From visual inspection of Subject 1, we can observe that her/his features suffered an alteration in phase two and phase four. These alterations indicate that the individual’s physiological variables suffered an alteration in the face of a mental challenge. These variabilities show that the baseline is different from the phases where the mental stressor is induced. Compared to the previous phase (phase four), the signals at the last phase are partially restored. We can conclude that the general behavior is similar to the baseline, specifically, the electromyography, skin conductance, breathing rate, and blood volume pulse. On the contrary, Subject 2, who represents an individual with a negative RSI, has physiological features in the last phase (recovery phase) similar to the stress phases (phases two and four). This indicates that the physiological features were not restored.

As shown in Section 3.5, we can visually inspect the physiological variables of a person in a two-dimensional space using PCA. Based on this, Figure 8 plots in two dimensions, variables from Figure 7. Figure 8a shows that for the first subject, the cluster that encompasses phase five is closer to the baseline than it is to phase four, and that translates into a positive RSI. On the contrary, Figure 8b shows that phase five is farther from the baseline for the second subject than it is to phase four, which results in a negative RSI.

Subject 1 has an RSI = 1.0, and an AF = 0.64, whereas Subject 2 has an RSI = −0.77, and an AF = 0.41. RSI equal to 1.0 means that Subject 1 has the highest value in our population. For Subject 2, phase five is far from the baseline, resulting in a negative RSI. When analyzing the AF values, for Subject 1, phase four is farther from the baseline than the same phase for the second subject, meaning that, in the first case, the variables suffered a bigger change in their values; that is, they showed a higher alteration factor (0.64>0.41).

Our results are consistent with the work of Healey and Picard [3]. The observed physiological features reflect the alteration that people suffer when induced by stressful situations; in this case, stress is induced by analytical challenges. Moreover, Lu, Wang, and You [15] suggest that there are traits of stress resilience that can be measured based on physiological responses. Our findings suggest that the five measured physiological features have enough information to capture an individual response to stress and that the severity is defined by how much these variables are altered.

## 5. Results Validation

This section explores the validation of the RSI using two procedures. The first one evaluates the RSI with the RESI-M tool. Even though there is no direct measure of resilience to stress, we could get some insights from comparing the index ranking against the RESI-M ranking. The second procedure evaluates the correlation of the raw physiological responses for each phase, to assess if the response has positive or negative resilience to stress and its quantification.

### 5.1. RESI-M Tool

The RESI-M scores range between 43 and 172, where a higher score means higher traits of resilience. By calculating the correlation between the average rank of the RESI-M scores and the RSI results, we compare the degree of correlation between the proposed RSI and the evaluation of the RESI-M scores. Table 3 shows the calculation of the RSI and RESI-M scores of seven individuals obtained with the four proposed methods. Additionally, it shows the average ranking for the seven individuals. From this point onwards, we refer to the RSI methods with the following abbreviations: Euclidean distance of principal components as ED-PCA, Mahalanobis distance as MD, cluster validity index distance as CVID, and Euclidean distance of kernel principal components as ED-KPCA.

To obtain the correlation coefficient, we first need to check for the distribution of the variables. If the variables are normally distributed, then the Pearson coefficient is calculated; otherwise, we obtain the Spearman coefficient. The Anderson-Darling test (AD) for normality is used to prove if the variables have normal distribution. From the results of the Anderson-Darling test with a significance level of 0.05, we can assume that RESI-M (AD = 0.455, *p*-value = 0.182), ED-PCA (AD = 0.155, *p*-value = 0.919), and CVID (AD = 0.658, *p*-value = 0.049) populations are normal. However, we can assume that MD (AD = 1.057, *p*-value < 0.005) and ED-KPCA (AD = 8.6616, *p*-value < 0.005) are not. Hence, we obtained the Pearson coefficient for ED-PCA and CVID, and the Spearman coefficient for MD and ED-KPCA.

The resulting correlation coefficient for the average ranking of the methods with a significance level of 0.10 was ED-PCA = −0.1149 (*p*-value = 0.801), MD = −0.714 (*p*-value = 0.071), CVID = −0.04 (*p*-value = 0.932), and ED-KPCA = −0.03571 (*p*-value = 0.939). These results demonstrate a high negative linear relationship between the results of the RESI-M and the MD method. Moreover, they show a low negative linear relationship between RESI-M and ED-PCA, and almost a null negative linear relationship with CVID and RESI-M, and ED-KPCA, and RESI-M. Furthermore, these latter associations would not be considered statistically significant.

A lower correlation value can indicate that the RSI measures a subset of the resilience whole spectrum. It also follows our previous statement that identifies resilience to stress as a subset of the resilience set. This is the case of the ED-PCA method, where it was expected to have lower correlation values since there is no direct method to measure resilience to stress based on physiological responses.

The limitation of this validation is the number of analyzed subjects of the RESI-M. Due to contact restrictions of the COVID-19 pandemic, the number of participants for the RESI-M test was low. Another limitation is the validation of the RESI-M tool. Even though it is validated in a Mexican sample, more subjects are needed to obtain statistically significant results. As we do not have a ground truth for this problem, a low correlation level was expected. That was the case with the ED-PCA, CVID, and ED-KPCA methods. However, the MD showed higher association traits to the RESI-M, which suggests that this method lines up better with the RESI-M test.

### 5.2. Correlation between Phase One and Phase Five

The second validation method involves capturing through the Spearman correlation if there is a monotonic relationship with its degree between phases one and five and determine if the behavior is resilient or non-resilient.

We calculated the Spearman coefficient of the seven test subjects for the five physiological features. Table 4 shows these results along with the mean of each subjects’ physiological variables and its corresponding result of the RESI-M test. The correlation coefficient was calculated with the preprocessed data of phase one and phase five. A sampling frequency of one every second was calculated with the average data of one second. In this case, we wanted to compare the baseline with the recovery phase to capture the relationship between both phases instead of the last stressor phase.

Next, we applied a Spearman correlation with the mean results of the seven subjects with its corresponding score of the RESI-M. We obtained a correlation of −0.50 (*p*-value = 0.253). These results conclude that we could not reject the null hypothesis of the Spearman coefficient, which states that there is no monotonic association between the mean results of the five physiological features and the RESI-M test.

In conclusion, even though there is no monotonic relationship between the mean results of the five physiological variables and the RESI-M, we have a significant relationship between the MD method and the RESI-M, proving a direct association of the first validation method of this analysis and the RESI-M.

### 5.3. Discussion

In the face of analytical challenges, the development of a Resilience to Stress Index was done using modern techniques of machine learning. Based on the Friedman results, we can infer that the most appropriate method compared to the proposed four to calculate the RSI is the Mahalanobis distance as there is no significant difference between the other three methods and its high monotonic association with the RESI-M. Even though the method requires high computational expense, it is recommended based on its relationship with the RESI-M test.

## 6. Conclusions

Even though measuring stress has been previously studied [3,15] with advanced statistics and machine learning tools, few studies, such as the one from Ćosić et al. [7], consider the evaluation of resilience to stress based on physiological response in individuals. Recent studies [8,9,10] have demonstrated the importance of resilience to stress for mental health. They are based on the idea that depending on the degree that individuals improve their resilience to stress, they will be more capable of coping with difficult situations and, as a result, be able to reduce their risk for depression or other mental disorders.

In this work, we showed that resilience to stress based on physiological response of a person can be quantified relative to a sample of individuals. As a result, we propose a new metric that can be measured and managed, namely a Resilience to Stress Index (RSI) along with an Alteration Factor (AF). Physiological features such as electromyography [30,31], breathing rate [26,27,28], blood volume pulse [22,23,24], skin conductivity [17,18], and peripheral temperature [20] can reflect stress conditions. In this work, the non-supervised technique of PCA enabled the visualization of five clusters reflecting the behavior of the physiological features during different mental stress test phases. The visual inspection of the data behavior allowed us to measure cluster distances between such phases. We measured these alterations with the Mahalanobis distance and the Euclidean distance in a space without multicollinearity. These *n*-dimensional space clusters reflect the general behavior of a subject towards stress when compared to a baseline.

The proposal of an RSI and AF serves as a solid foundation for continuing research within resilience to stress and psychiatric studies. The benefits of having a metric for measuring resilience to stress are multiple. For instance, to the extent that individuals can track their resilience to stress, they can improve their ordinary life. The proposed index can be measured after and before clinical treatment to assess any improvement. Embedded sensors in a wearable are suitable to obtain additional features that capture different behaviors; hence, the proposed methodology can be improved with data from these non-invasive devices. Additionally, several tests can be applied to measure the severity of various psychophysiological responses, such as depression.

The methodology presented in this work can be replicated using a similar biofeedback device and psychophysiological stress test to collect the specified variables and serve as a baseline to further research on resilience to stress of individuals. Additionally, it can contribute to the study of depression and other mental disorders with the aim to offer health organizations the tools to identify and support individuals with higher risks of depression and, most importantly, prevent any risk to their physical integrity.

In comparison to related work, this study proposed four robust RSIs independent of socio-demographic variables, as they are calculated based on relative distances of each subject and independent of bias inherent in written tests. The limitations of this study assume that the population has healthy biological parameters, that an individual cannot relax more than he was at the beginning of the psychophysiological stress test, and that the relative comparison between samples depends on the sample parameters.

Future work aims to validate the robustness of the RSI index, by applying our methodology to a different group of subjects, under the assumption that resilience to stress should be different among distinct populations. This group is foreseen to include different nationalities, geographical locations, and social-economic conditions. Additionally, we will explore if all five physiological features are needed for the RSI and AF calculations, or if it is possible to reduce this number to the most relevant ones without compromising the quality of the new metrics.

## Figures and Tables

**Figure 1 sensors-21-08293-f001:**
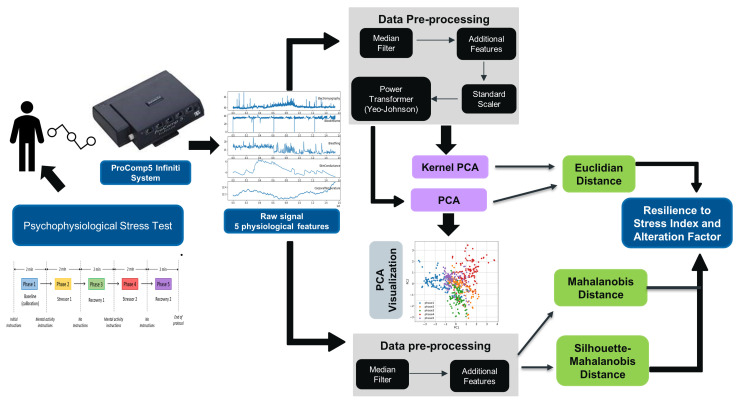
Flowchart that shows the main steps of the methodology to compute the new metrics. The ProComp5 image is taken from the ProComp5 Infiniti${}^{\mathrm{TM}}$ Hardware Manual with the consent of Thought Technology, Montreal, QC, Canada. www.thoughttechnology.com (accessed on 6 December 2021).

**Figure 2 sensors-21-08293-f002:**
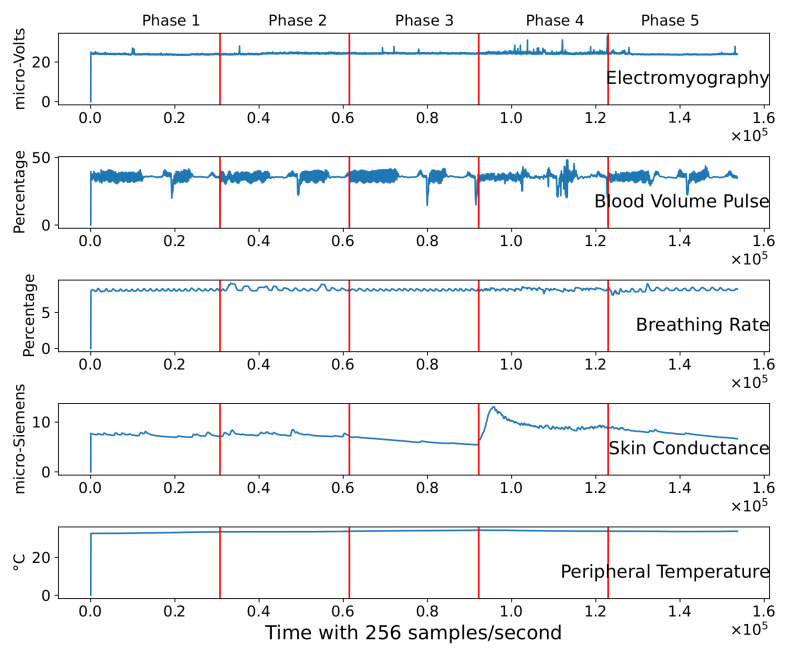
Raw signal of the five physiological features of one subject during the psychophysiological stress test.

**Figure 3 sensors-21-08293-f003:**
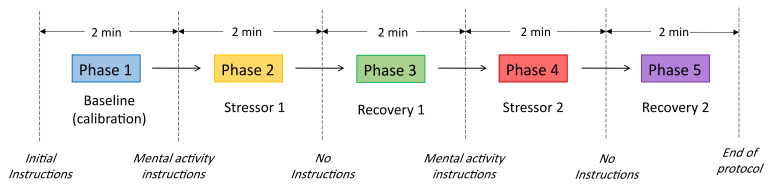
Flow diagram of the psychophysiological stress protocol.

**Figure 4 sensors-21-08293-f004:**
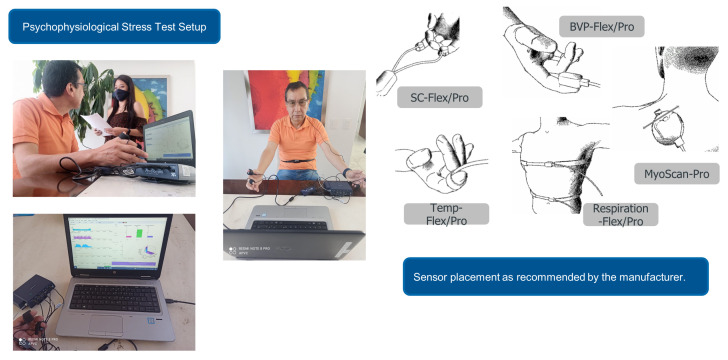
Stress test setup with connected subject on the left, and on the right, sensor placement as recommended by the manufacturer. Images are taken from the ProComp5 Infiniti${}^{\mathrm{TM}}$ Hardware Manual with the consent of Thought Technology, Montreal, QC, Canada. www.thoughttechnology.com (accessed on 6 December 2021).

**Figure 5 sensors-21-08293-f005:**
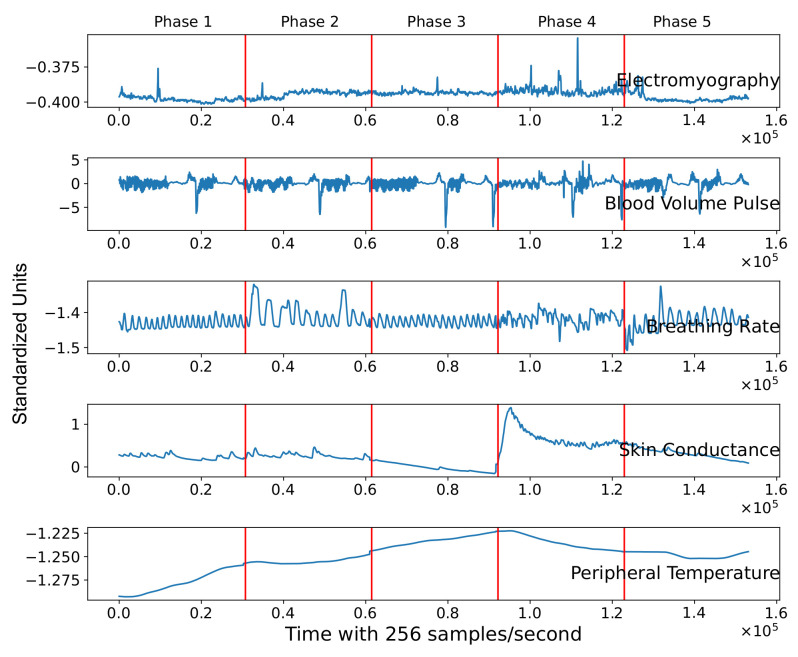
Filtered and standardized signals of the five physiological features are divided by the psychophysiological stress test phases.

**Figure 6 sensors-21-08293-f006:**
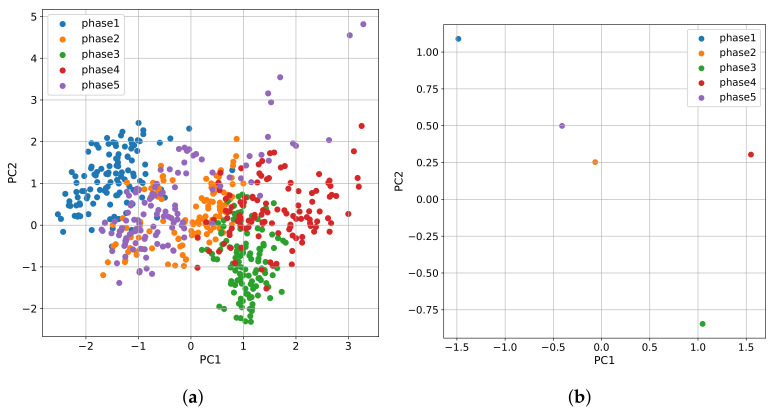
Two-dimensional space of PCA from a subject (**a**) and its corresponding centroids (**b**).

**Figure 7 sensors-21-08293-f007:**
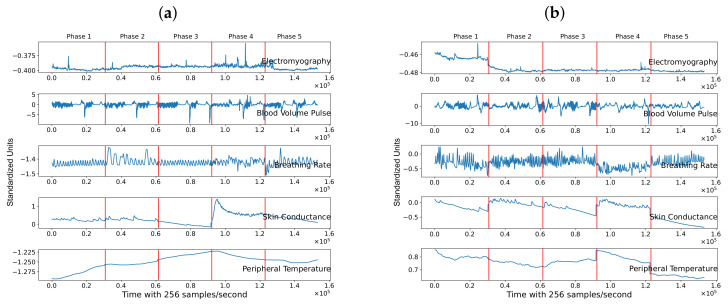
Standardized physiological features of two different subjects. Subject 1 depicted in (**a**), represents an individual with positive RSI, and Subject 2 shown in (**b**), represents an individual with negative RSI.

**Figure 8 sensors-21-08293-f008:**
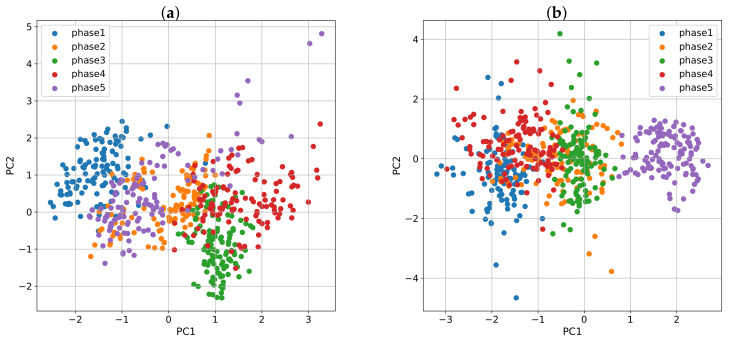
2D-PCA representation of two different subjects. Subject 1 depicted in (**a**), is an individual with a positive RSI, and Subject 2 in (**b**), an individual with negative RSI.

**Table 1 sensors-21-08293-t001:** Description of variables measured with the biofeedback device.

	Skin Conductance (SC)	Blood Volume Pulse (BVP)	Peripheral Temperature (PT)	Electromyography (EMG)	Breathing Rate (BR)
Description	Measures the peripheral skin conductance	Measures changes in the arterial translucency	Measures changes in corporal temperature	Measures muscle response	Measures the expansion and contraction of the rib cage
Sensor	SC-Flex/Pro	BVP-Flex/Pro	Temp-Flex/Pro	MyoScan-Pro	Respiration-Flex/Pro
Unit of Measure	micro-Siemens	Unit less quantity displayed as percentage	Centigrade Degrees	Volts	Unit less quantity displayed as percentage
Range of values	0–30 μS	0–100%	10–45 ${}^{\circ }$C	0–1600 μV	0–100%
Typical Relaxed Measures	2 μS	30–60%	18 ${}^{\circ }$C	3–5 μV	No typical waveform

**Table 2 sensors-21-08293-t002:** Median, median absolute deviation, and mean rank results of the Friedman test.

Metric	MD	MAD	MR
Euclidean of Principal Components	0.077 ± 0.083	0.183	2.183
Mahalanobis	0.174 ± 0.119	0.278	1.831
Silhouette with Mahalanobis	0.030 ± 0.083	0.183	2.183
Euclidean of Kernel PCA	0.077 ± 0.308	0.416	2.465

**Table 3 sensors-21-08293-t003:** RSI-RESI-M results and average ranking of seven individuals using the four proposed methods. Table is sorted by RESI-M rank.

RSI BASED ON				
**RESI-M Test (Rank)**	**ED-PCA (Rank)**	**MD (Rank)**	**CVID (Rank)**	**ED-KPCA (Rank)**
106 (1)	0.2175 (4)	0.2420 (6)	0.0137 (4)	0.0813 (7)
138 (2)	0.5728 (7)	0.2562 (7)	0.6317 (6)	0.0075 (3)
142 (3)	−0.0133 (1)	0.2024 (3)	0.1203 (5)	−0.0090 (2)
148 (4)	0.2853 (5)	0.2249 (5)	1.0000 (7)	0.0232 (4)
152 (5)	0.4129 (6)	−0.5933 (1)	−0.1771 (1)	−0.0151 (1)
159 (6)	0.0998 (2)	0.2200 (4)	−0.0230 (2)	0.0490 (6)
162 (7)	0.1633 (3)	−0.1718 (2)	0.0081 (3)	0.0318 (5)

**Table 4 sensors-21-08293-t004:** Spearman correlation coefficient of the seven test subjects physiological variables with its corresponding RESI-M test result.

Subject ID	EMG	HR	BR	SC	PT	Mean	RESI-M Test
1	−0.45	0.25	−0.27	0.92	−0.99	−0.11	159
2	0.00	0.27	0.11	0.99	0.87	0.45	148
3	−0.03	−0.09	−0.02	0.88	0.94	0.34	138
4	−0.31	0.08	0.16	0.96	−0.35	0.11	152
5	−0.09	0.24	0.14	0.99	−0.82	0.09	106
6	−0.10	0.15	0.02	0.99	−1.00	0.01	142
7	−0.11	0.04	0.03	0.99	−1.00	−0.01	162

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
