# Peer review of "Towards a Resilience to Stress Index Based on Physiological Response: A Machine Learning Approach"

_sensors, 2021, doi:10.3390/s21248293_

Round 1

Reviewer 1 Report

The manuscript entitled: "Towards a Resilience to Stress Index Based on Physiological Response: A Machine Learning Approach." This study proposes a new index to measure an individual's resilience to stress based on the changes of specific physiological variables. These variables include electromyography, the muscle response, blood volume pulse, breathing rate, peripheral temperature, and skin conductance. The study measured the data with a biofeedback device from individuals subjected to a 10-minutes psychophysiological stress test. The data exploration revealed that features' variability among test phases could be observed in a two-dimensional space with Principal Components. Analysis (PCA). In this work, the study demonstrates that each feature's values within a phase are well organized in clusters. Based on this observation, the study proposes the new index, Resilience to Stress Index (RSI). To compute the index, the study used non-supervised machine learning methods to calculate the inter-cluster distances, explicitly using the following four methods: Euclidean Distance of PCA, Mahalanobis Distance, Cluster Validity Index Distance, and Euclidean Distance of Kernel PCA. While there was no statistically significant difference (p > 0.01) among the methods, the study recommends using Mahalanobis since this method provides a higher monotonic association with the Resilience in Mexicans (RESI-M) scale. Results are encouraging since the study demonstrated that the computation of a reliable RSI is possible.  

The manuscript has the following contributions:

  1.  The study identified that the expected variables to increase when an individual is exposed to stress are: electromyography, skin conductance, and breathing rate. Moreover, the variables expected to decrease are peripheral temperature and heart rate variability. Given this, the time a person takes to return its heart rate variability, respiratory rate, muscle tension, temperature, and skin conductance to its baseline parameters can help us define the resilience to a person's stress.
  2. The study presented efficient solutions to the considered problem.
  3. The formulation of the problem is suitable and correct.
  4. The simulation results showed the optimal performances of the proposed schemes. 

However, I have the following suggestions to improve the manuscript optimally.

  1. There should be a notations table in the problem description part.
  2. The flowchart of the proposed scheme must be there in the manuscript.
  3. Add the time complexity of the machine learning algorithms in the study.
  4. Add the statistical method on which the experiment was conducted in the study.
  5. Add the finding and limitation subsection of the study before the conclusion. 
  6. Write the future work of the study in the conclusion part. 

Reviewer 3 Report

The authors present research towards the computation of a new resilience to stress index based on physiological features by using non-supervised machine learning methods. They identified that there was no such quantifiable index so far and defined an appropriate method to quantify the alteration intensity of physiological features by proposing a new factor. Furthermore, the authors validate the correctness of the new index by comparing it to an established resilience scale. Although the validation is limited, the results and methods can serve as a base to further research on resilience to stress. There are some issues that need to be addressed to further improve the quality of the paper:

  1. Introduction:
    • HRV is a very important physiological feature related to stress. Why is it not used by the authors and only mentioned in one sentence? The authors should explain better their opinion on HRV.
    • The authors should use the term blood pressure (BP) instead of blood volume pressure because the abbreviation BVP is generally used for blood volume pulse which is related to PPG and thus to HRV.
    • Maybe section 1.1.5. should be about HRV or blood volume pulse instead of blood pressure???
  2. Methodology:
    • Line 230: I assume that “peripheral temperature” should be “skin conductance”
    • Table 1: please check sensor name of SC; please check unit of BVP
    • Could you give some information on the mathematical and language tasks? It is not clear what happens in the stressor phases
    • Please check section 3.2 if you indicate the sensor location that you used in the setup, e.g. are EMG electrodes placed on the trapezius muscle?
    • A photo of the test setup with connected subject would be interesting to see
    • Could you comment on the feature normalization: why is there still a unit given in Fig. 3?
    • Section 3.6.5 could be omitted
  3. Results:
    • The results for the methods described in 3.6 to calculate the distance of phase one against all other phases are not mentioned. Could you comment on this?
  4. Discussion:
    • Please add references to the text
    • Could you comment to the question if all the five physiological features are necessary for your method or if you could reduce the number of features to the most relevant ones without compromising the quality of the results?
    • Despite the novel approach and the promising results, application of the technology seems to be limited. Primary focus is on stationary use such as in a lab or clinical setting when the subject is in a resting position connected to many sensors/electrodes. Could you comment on the potential use with wearable/portable devices taking into account lower biosignal sampling and reduced computing power as well as reduced number of sensors/physiological features. It is unlikely that an individual would track its resilience to stress in everyday life with all the sensors used in this study.

Round 2

Reviewer 1 Report

The current version of the manuscript is satisfactory, there I have no comments more. However, still, English can be improved by authors before final publications in the journal.

Reviewer 3 Report

Thank you very much for your detailed revision.